# Cortical Location of Language Function May Differ between Languages While White Matter Pathways Are Similar in Brain Lesion Patients

**DOI:** 10.3390/brainsci13081141

**Published:** 2023-07-29

**Authors:** Corinna Boerner, Axel Schroeder, Bernhard Meyer, Sandro M. Krieg, Sebastian Ille

**Affiliations:** 1Department of Neurosurgery, Klinikum rechts der Isar, School of Medicine, Technical University Munich, 81675 Munich, Germany; 2TUM-Neuroimaging Center, Klinikum rechts der Isar, School of Medicine, Technical University Munich, 81675 Munich, Germany

**Keywords:** bilingual, glioma, language, tractography, transcranial magnetic stimulation

## Abstract

The neural representation of language can be identified cortically using navigated repetitive transcranial magnetic stimulation and subcortically using the fiber tracking of diffusion tensor imaging. We investigated how cortical locations of language and language-eloquent white matter pathways differ in 40 brain lesion patients speaking various languages. Error rates related to stimulations at single sites in the frontal and parietal lobe differed significantly between Balto-Slavic and Indo-European languages. Error rates related to stimulations at single sites in the temporal lobe differed significantly between bilingual individuals. No differences were found in the white matter language pathway volumes between Balto-Slavic and Indo-European languages nor between bilingual patients. These original and exploratory data indicate that the underlying subcortical structure might be similar across languages, with initially observed differences in the cortical location of language depending on the semantic processing, but these could not be confirmed using detailed statistical analyses pointing at a similar cortical and subcortical network.

## 1. Introduction

The neural representation of language is complex and highly variable across individuals. Recent models of the language network indicate that cortical and subcortical levels underlie the phonological and semantic processing of language [1,2,3,4]. Dual-stream models of language distinguish between dorsal streams, mainly responsible for phonological processing and speech articulation, and ventral streams, which are thought to play a role in the processing of semantic information [1,2,3,5]. 

On the subcortical level, ventral stream I includes the extreme fiber capsule (which the uncinate fascicle and the inferior fronto-occipital fascicle are part of) connecting Brodmann’s area 45 and the temporal cortex. Ventral stream II includes the uncinate fascicle connecting the frontal operculum and the superior temporal gyrus. Dorsal stream I includes the arcuate fascicle and the superior longitudinal fascicle connecting the superior temporal gyrus (STG) and the premotor cortex. In parallel, dorsal stream II connects the superior temporal gyrus and Brodmann’s area 44 [3,6,7].

On the cortical level, the inferior frontal gyrus, including Broca’s area; the superior temporal gyrus, including Wernicke’s area; the middle temporal gyrus; the inferior parietal gyrus; and the angular gyrus play a critical role in language processing [3,5,6].

In patients suffering from left-sided perisylvian brain lesions, a resection might lead to language impairment. Therefore, the aim is to maximize resection while preserving the patient’s language function and thereby the patient’s quality of life. The preservation of white matter pathways is of special importance since subcortical tracts have a limited potential for plasticity [8,9,10,11,12,13]. Cortical parts of the language networks, on the other hand, seem to be prone to plastic reorganization [14,15].

To obtain the optimal functional and oncological outcome of brain lesion resections, it is necessary to localize the patient’s individual language-eloquent brain regions [13,16,17,18]. The gold standard is the use of direct cortical stimulation (DCS) during awake craniotomy [17,19,20,21].

As an adjunct to DCS, navigated repetitive transcranial magnetic stimulation (nrTMS) can be used to localize individual language sites [22,23,24,25]. The combination of nrTMS and diffusion tensor imaging fiber trackings (DTI FTs) can be used to identify white matter fibers emanating from the cortical language sites, which may therefore be potentially involved in language [22,26,27,28]. To obtain an optimal outcome after the surgical resection of language-eloquent brain lesions, it is a common practice to use the language mappings of all fluently spoken languages. 

In the past, multiple studies non-invasively examined differences between cortical language-eloquent sites for more than one spoken language in healthy subjects using functional MRI (fMRI) and during awake craniotomy in patients using DCS [29,30,31,32,33,34,35,36]. Additionally, nrTMS has been used to identify the differences in cortical language localization in healthy subjects [37,38]. In summary, most of these studies were able to identify differences in cortical language localizations for two different spoken languages. 

The present study aims to preoperatively investigate the cortical location of language function using nrTMS language mapping and language-eloquent white matter pathways through nrTMS-based DTI FT in patients suffering from left-sided perisylvian brain lesions speaking various languages. More specifically, four language groups are compared: (1) Balto-Slavic languages, (2) all other Indo-European languages (except for Slavic languages, later referred to only as Indo-European languages), (3) Arabic, and (4) Chinese, as well as various languages of bilingual individuals. We hypothesize that the localization of language function differs between various languages [39].

## 2. Methods

### 2.1. Patients

This study included 40 patients from our cohort of patients whose language function was mapped between 2014 and 2019. Included patients had a left-sided perisylvian brain lesion, were right-handed according to the Edinburgh Handedness Inventory [40], and were preoperatively mapped using an object-naming task in their respective languages. Exclusion criteria were (1) general MRI and nrTMS exclusion criteria (e.g., cochlear implants or cardiac pacemakers) [41], (2) age below 18 years, (3) too-severe aphasia [42], and (4) missing data. From the initial 57 patients, 17 were excluded due to the abovementioned reasons (too severe aphasia *n* = 5; missing data *n* = 12) leading to a final sample size of 40 patients. Detailed patient characteristics can be found in Table 1. Patients were divided into Balto-Slavic and Indo-European groups as well as a group of bilingual patients (Table 2 and Table 3). The status of true bilingualism, meaning the learning of both languages before the age of 10, could not be verified in all patients who were included in the group of bilinguals in the present study [43]. While a definition of “true” bilingualism is difficult, as discussion by experts in this field of research has shown, we decided to include unverified patients in the bilingual group speaking two languages to investigate similarities and differences in language function in these patients undergoing language-eloquent brain tumor resection, which has relevant clinical implications.

Patients gave written informed consent prior to the examination. The study was approved by the local ethics committee and was performed in accordance with the Declaration of Helsinki.

### 2.2. Magnetic Resonance Imaging and Diffusion Tensor Imaging

Magnetic resonance imaging was performed according to the standard protocol at our department. Images included a T1-weighted 3D gradient echo sequence with intravenous contrast administration, a T2-weighted 3D FLAIR sequence, and DTI sequences with 32 orthogonal sequences. All sequences were performed on 3T magnetic resonance scanners (Achieva 3T, Philips Medical System, Best, The Netherlands, B.V.). 

### 2.3. Cortical Analysis: nrTMS Language Mapping

The nrTMS language mappings were performed using the eXimia nTMS system version 4.0/5.0 and a NEXSPEECH^®^ module (Nexstim Plc, Helsinki, Finland). The T1-weighted MRI sequence was used for the co-registration with the patient’s head. After the resting motor threshold was identified, the patient performed object naming [42]. The object-naming task used in the present cohort is implemented within the nTMS software and used by multiple users [25,42,44,45,46,47]. Furthermore, the object-naming task is, among others, still standardly used for intraoperative procedures by multiple departments [1,48,49,50]. Thus, the object-naming task is repeatedly used for language testing in different languages. First, two baseline measurements were run without nrTMS stimulation to ensure that only images that could be correctly named by the patient were included. The inter-picture interval (IPI) was set between 2500 ms and 3500 ms, and the display time (DT) was set between 700 ms and 2000 ms. Next, the correctly named images were presented, time-locked to nrTMS pulses (5 pulses at 5 Hz, picture-trigger-interval (PTI) = 0 ms) with an intensity of 100% of the resting motor threshold. The stimulations were administered to 46 predefined cortical stimulation sites on the left hemisphere, with every site being stimulated three times [22,51,52]. Detailed mapping characteristics can be found in Table 1. Baseline and stimulation measurements were video-recorded to analyze naming errors by comparing the stimulation trial with the baseline measurements. Naming errors were categorized into no-response errors, performance errors, semantic errors, phonological errors, and neologisms [42,52]. Language-positive cortical sites in terms of nrTMS were identified as brain regions, at which naming errors were elicited by the nrTMS stimulation.

### 2.4. Subcortical Analysis: nrTMS-Based DTI Pathway Tractography

The steps of the DTI FT are shown in Figure 1. First, left-sided language-positive sites identified using nrTMS were exported to perform the nrTMS-based DTI FT tractography. DTI FT was performed using our standard deterministic algorithm with fiber assignment via continuous tracking (FACT) (iPlanNet Cranial 3.0, Brainlab AG, Munich, Germany) and according to the protocol of Negwer and colleagues [26,27,53]. A 5 mm rim was added around the language-positive cortical sites, which were then set as regions of interest (ROIs) for the tractography. Next, the entire subcortical language network emerging from the nrTMS-language-positive sites was calculated using the software. In addition to the language-positive site ROIs, a second ROI was manually added in order to visualize single pre-defined language-related pathways. In particular, the inferior fronto-occipital fascicle (IFOF) was created by adding an ROI in the anterior–posterior-oriented fibers of the external capsule. The superior longitudinal and arcuate fascicle (SLF/AF) were visualized using an ROI in the anterior–posterior-oriented fibers lateral to the posterior horn of the lateral ventricle. Finally, the frontal aslant tract (FAT) was tracked with an ROI in the cranial-caudal oriented fibers connecting the superior and inferior frontal gyri [54]. All tractographies were performed with a minimum fiber length of 100 mm and a fractional anisotropy (FA) of 0.10 or 0.15 since these parameters were shown to lead to an optimal visualization of white matter language pathways [27]. For language tractography of bilingual individuals, the optimal FA parameters were defined for one language and then applied to the second language in order to perform a direct comparison.

### 2.5. Data Analysis

For the cortical analysis, error rates were calculated by dividing the number of errors made at a certain stimulation site (as defined by the CPS) [51,52] by the total number of stimulations at that site. Averaging these error rates across all subjects resulted in a single average error rate score for each of the 46 stimulation sites. For the visualization of the error rate distribution, we created heat maps and divided the error rates into six categories: (1) <4% (grey), (2) 4–8% (light yellow), (3) 8–12% (dark yellow), (4) 12–16% (orange), (5) 16–19% (light red), and (6) >20% (dark red). In order to test whether there were differences in error rates between Balto-Slavic (*n* = 24) and Indo-European (*n* = 25) languages, we ran Mann–Whitney U tests. Additionally, error rates were dichotomized into “error” and “no error”. We ran a generalized linear mixed model with the dichotomized error variable as a target, language and stimulation site as fixed factors, and study ID as a random factor. All statistical tests were performed in SPSS. Since the groups of Arabic (*n* = 4) and Chinese languages (*n* = 2) were too small for statistical analyses, we decided to perform an exploratory heat map analysis for these groups instead. In addition, we ran Wilcoxon Signed Rank Z tests to examine whether there were differences in error rates between the two spoken languages of bilingual individuals (*n* = 18). For the subcortical analysis, iPlanNet Cranial 3.0 (Brainlab AG, Munich, Germany) was used to calculate the volumes of the language network emerging from the nrTMS-language-positive ROIs and of the individual language-related pathways (IFOF, SLF/AF, FAT). In order to test whether there were differences in the volumes of the language network emerging from the nrTMS-language-positive ROIs and of the individual language-related pathways (IFOF, SLF/AF, FAT) between Balto-Slavic (*n* = 23) and Indo-European (*n* = 25) languages, we ran Mann–Whitney U tests. For the analysis of bilingual individuals, we ran Wilcoxon Signed Rank Z tests to examine whether there were differences between the two spoken languages of bilingual individuals in white matter pathway volumes (*n* = 19). The significance level was set to *α* = 0.05; results with a significance level between 0.05 and 0.10 were marked as marginally significant. All analyses were corrected for multiple comparisons using the Benjamini–Hochberg procedure. Effect sizes of significant differences were calculated using the following formula: $\eta^{2}=\frac{Z^{2}}{N-1}$. In addition, we calculated the Spearman’s correlations between the error rates and relative white matter pathway volumes for Balto-Slavic (*n* = 23) and Indo-European languages (*n* = 25).

## 3. Results

### 3.1. Patient and Mapping Characteristics

Of the 40 patients included in this study, 25 were male (62.50%) and 15 were female (37.50%). The mean age was 43.70 ± 14.58 years (mean ± standard deviation). Mappings were performed using a mean resting motor threshold of 33.81 ± 7.69% stimulator output (minimum: 17%, maximum: 59%) and a mapping intensity of 100% resting motor threshold. Age, gender, handedness, and the resting motor threshold did not significantly differ between the Balto-Slavic and the Indo-European language groups. Overall, nrTMS elicited a mean of 17.60 ± 14.20 naming errors during the stimulation, of which 10.49 ± 10.98 were no-response errors, 4.35 ± 5.97 were performance errors, 2.48 ± 3.45 were semantic errors, 0.16 ± 0.44 were neologisms, and 0.11 ± 0.36 were phonological errors.

The following case numbers are included in each language group and in each analysis type: The error rates were analyzed for 24 Balto-Slavic, 25 Indo-European, 4 Arabic, 2 Chinese, and 18 bilingual patients. The DTI FT was analyzed for 23 Balto-Slavic, 25 Indo-European, and 19 bilingual patients (Table 4). 

### 3.2. Cortical Analysis: Error Rates

Figure 2 displays the mean error rates for each stimulation site as heat maps. An exploratory analysis of the heat maps indicates that the highest error rates were related to stimulations in the posterior middle frontal gyrus and the posterior superior temporal gyrus for Balto-Slavic languages and in the middle postcentral gyrus for Indo-European languages. With some reservations, because of the low number of patients included, the highest error rates were found within the superior and middle frontal gyrus, as well as the postcentral gyrus for the Arabic language, and in the superior, middle, and inferior frontal gyrus as well as the angular gyrus for the Chinese language (Figure 2).

Mann–Whitney U tests showed that the languages significantly differed in error rates related to stimulations in the middle superior frontal gyrus (*U* = 213.5, *N*_1_ = 24, *N*_2_ = 25, *p* = 0.026, *ƞ*^2^ = 0.104), the posterior middle frontal gyrus (*U* = 211.5, *N*_1_ = 24, *N*_2_ = 25, *p* = 0.046, *ƞ*^2^ = 0.083), and the middle postcentral gyrus (*U* = 222, *N*_1_ = 24, *N*_2_ = 25, *p* = 0.027, *ƞ*^2^ = 0.102), with Balto-Slavic patients showing more errors than Indo-European patients (Appendix A, Figure 3). After correction for multiple testing, no significant differences were found. Furthermore, analyses using a generalized linear mixed model indicated no significant influence of language (*p* = 1.000) or stimulation site (*p* = 0.807) on the presence or absence of errors (Appendix A).

The results of the Wilcoxon Signed Rank tests show that the two languages spoken by bilingual individuals differed significantly in error rates related to stimulations in the posterior superior temporal gyrus (*Z* = −2.264, *p* = 0.024, *ƞ*^2^ = 0.302) and the posterior middle temporal gyrus (*Z* = −2.414, *p* = 0.016, *ƞ*^2^ = 0.343; Appendix A, Figure 3). After correction for multiple testing, no significant differences were found.

### 3.3. Subcortical Analysis: Pathway Tractography

The results of the Mann–Whitney U tests using the pathway volume based on an FA value of 0.10 showed that the SLF/AF differed significantly with Balto-Slavic subjects having a larger volume than Indo-European subjects (*U* = 189.00, *N*_1_ = 24, *N*_2_ = 25, *p* = 0.042, *ƞ*^2^ = 0.088; Table 5). However, this difference was not significant after correction for multiple testing. All other language-related white matter pathway volumes based on an FA of 0.10 and 0.15 did not differ between Balto-Slavic and Indo-European languages. When correcting for the total volume of the subcortical language network by dividing the volume of the individual pathways (i.e., IFOF, SLF/AF, and FAT) by the total volume of the network, all relative pathway volumes did not significantly differ between Balto-Slavic and Indo-European languages. In addition, the absolute and relative volumes of language-related white matter pathways did not significantly differ in bilingual individuals (Table 5).

### 3.4. Correlations between Error Rates and Pathway Volumes

We calculated the correlations between the error rates and the relative pathway volumes for the Balto-Slavic language group (Appendix A) and the Indo-European language group (Appendix A). All relative pathway volumes based on an FA value of 0.10 and 0.15 were used to determine the correlations. 

## 4. Discussion

### 4.1. Cortical Differences and Subcortical Similarity

This study compared the cortical and subcortical locations of language function between various languages. Our results show that the cortical location of language might differ between languages, while the subcortical location seems to be similar, or at least not significantly different, across languages. More specifically, error rates differ between Balto-Slavic and Indo-European languages, as well as between the two spoken languages in bilingual patients (Appendix A, Figure 2 and Figure 3), and differences are mainly found in known cortical language regions such as the anterior and posterior language-related regions. Regarding classic models of cortical language function, meaning the anterior located Broca’s area and the posterior located Wernicke’s and Geschwind’s area, the present data indicate that Balto-Slavic languages are more disseminated with high error rates in both anterior and posterior language-related regions. In contrast, high error rates for Indo-European languages could primarily be found within and adjacent to the classical Broca’s area (Figure 2). This difference is also reflected in differences between the languages’ error rates (Figure 3). Even though the differences in error rates were not significant after correction for multiple comparisons and analyses using a generalized linear mixed model indicated no significant influence of language or stimulation site on the presence or absence of errors, the results should still be considered, since the effect sizes are within the range of medium to large effects [55] (Appendix A). With this in mind, more detailed statistical analyses could not confirm cortical differences as observed in the pure comparison of error rates. 

In contrast, the absolute and relative volumes of language-related white matter pathways (IFOF, SLF/AF, FAT) did not differ between various languages (Table 5). The applied approach of function-based tractography using nrTMS-positive cortical stimulation sites as ROIs indicates no difference in the amount of recruited language tracts between the languages. Similarly, we found moderate correlations between the relative volumes of the IFOF, SLF/AF, and FAT and error rates related to stimulations in classic anterior and posterior language areas for both Balto-Slavic and Indo-European languages. Together, these results indicate that different languages seem to have a similar structure of language-related white matter pathways.

### 4.2. Concordance with Previous Literature

The present results, examined using nrTMS language mapping, correspond to most prior studies on cortical differences for two spoken languages in patients [30,31,32]. Furthermore, while recent studies using fMRI also showed a universal language network [56], our findings are in line with previous research examining the cortical localization of two spoken languages in healthy bilingual subjects [34,36,37,38]. For the left hemisphere, the researchers of these studies could also find high error rates related to stimulations in anterior and posterior language-related regions for one language and in different anterior and posterior sites for the other language [38]. Interestingly, the error rates differed significantly between the languages spoken by bilingual subjects on the left hemisphere for stimulations in the triangular part of the inferior frontal gyrus and the posterior middle temporal gyrus, and marginally significant differences were found for error rates related to stimulations in the angular gyrus and the superior parietal lobe [38].

These differences, among others, were also replicated in our study in the left hemisphere in bilingual patients. Together with the studies by Tussis et al. [37,38], this underlines the fact that the two spoken languages in bilingual individuals are localized in language-specific brain regions [30,31,32], even though some brain regions might process language in general (i.e., these areas are shared by both languages) [29,31,32]. The fact that languages are localized in different brain regions is also supported by a case study indicating that bilingual patients with brain lesions might show impairments in one language but not the other [57].

Moreover, our results are in accordance with the reported plasticity in different brain regions: while cortical regions show plasticity [14,15], white matter pathways have limited capacity for plasticity [8,9,10,11,12,58,59]. Our results show further evidence that cortical regions are able to functionally reorganize since the language function can have different anatomical cortical locations depending on the language. Language-related white matter pathways, on the other hand, seem to be similar, which might be related to the limited plasticity of subcortical structures [8,9,10,11,12,58,59].

### 4.3. Limitations

A limitation of this study is that the Balto-Slavic and Indo-European language groups were composed of patients speaking various languages, and this heterogeneity might have influenced our group comparisons. In addition, languages can be grouped according to several linguistic theories/models, and up to 406 language families were recently identified [4], which complicated language grouping in our patient cohort. In particular, the Indo-European group comprises languages with different origins (also including one Finnish and one Hungarian case). This has to be considered when interpreting the results, since Finno-Urgic languages may have formed their own group. Moreover, the classification of Turkish using standard systems and the discussion on the classification of Turkish is controversial. Turkish is not standardly part of Indo-European languages, but there is still discussion on its origin [60]. However, after linguistic consultation on similarities, this grouping was used not least to gain a decent sample size for statistical comparisons. Moreover, future research should investigate whether the variability in language localization between individuals speaking a similar language is lower than the variability within bilingual patients.

Furthermore, although the object-naming task used in the present cohort is used by multiple users [25,42,44,45,46,47] and, among others, still standardly used for intraoperative procedures by multiple departments [48,49,50], there is no standardization of the task, nor for perioperative test materials for all languages of patients included in the present study. This fact might particularly impact the number of false positives. In addition, the object-naming task is a clinical tool that only includes the naming of nouns. From a linguistic perspective, the inclusion of verbs, syntax, and compositional semantics would give a better indication of language function. Moreover, one default picture dataset was used for the object-naming task, which was initially developed for the German language. Thus, differences in the frequency or phonological characteristics of the words in different languages might have confounded the results. However, data were collected during pre-surgical planning, including brain lesion patients who had language deficits, which is why a more complicated language task was not feasible.

Cortical differences between Balto-Slavic and Indo-European language groups were found in superior and middle frontal areas, which process not only language function but also cognitive control. Thus, reported differences might be influenced by a cognitive control effect, which must be considered when interpreting the results.

The fact that this study included patients with brain lesions implies that cortical plasticity can occur merely due to the lesion itself. Moreover, a brain lesion can induce tissue displacement, which might bias the subcortical tractography. This should be taken into account when interpreting the findings. In addition, subcortical tractography was analyzed based on DTI FT and the limitations of DTI, e.g., biases due to crossing fibers need to be considered when interpreting the results. Although DTI sequences with 32 directions and a deterministic algorithm of DTI FT were used to minimize this bias as much as possible, limitations of tractography per se and the fact that it can only give an impression of subcortical white matter pathways and being used for scientific comparisons in this study without showing the true subcortical anatomy, must be highlighted. 

Lastly, data were acquired during clinical routine and analyzed retrospectively, which is why additional information on bilingual patients (e.g., age of language acquisition, language proficiency) cannot be given.

Moreover, comparative analyses on equivalent age- and gender-matched populations were not performed in this study. To further elucidate such differences, analysis of a several-fold larger cohort that is required. Yet we hope that the current study provides valuable pilot data for this endeavor.

## 5. Conclusions

The present results indicate that the underlying subcortical structure might be similar across languages, and we initially found differences in error rates in the cortical locations of language between various languages, as examined using non-invasive nrTMS language mapping and nrTMS-based DTI FT. However, differences in error rates were not significant after correction for multiple comparisons, and analyses using a generalized linear mixed model indicated no significant influence of language or stimulation site on the presence or absence of errors. Hence, these detailed statistical analyses could not confirm cortical differences between languages as observed by the pure error rate analysis pointing at a similar cortical and subcortical network. Nevertheless, these first data on non-invasive mappings in patients with left-sided perisylvian brain lesions speaking various languages should only be used to discuss trends and hypotheses of the language network. In contrast, the present results do not allow us to make final conclusions on the language localization in this cohort of patients or to make clinical decisions. Future prospective studies must include larger cohorts with respect to language clustering and the limitations of the present study. Additionally, intraoperative data and connectome analyses have to be added to investigate the language network comprehensively.

## Figures and Tables

**Figure 1 brainsci-13-01141-f001:**
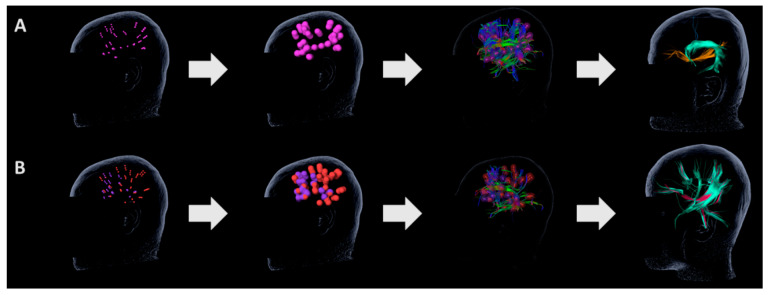
**Summary of the nrTMS-based DTI pathway tractography methodology.** (**A**) nrTMS-based positive language sites were used as regions of interest (ROIs). Next, a 5 mm rim was added around the ROIs. Using the ROIs, the whole language network was tracked. After the whole language network was tracked, additional ROIs were added in order to visualize the following single pathways: inferior fronto-occipital fascicle (IFOF, orange), superior longitudinal and arcuate fascicle (SLF/AF, green), and frontal aslant tract (FAT, blue). (**B**) A similar methodology was used for bilingual subjects. nrTMS-based positive language sites of two languages (red and pink) were used as ROIs. The whole language network could be tracked for each spoken language (language 1 in red, language 2 in green).

**Figure 2 brainsci-13-01141-f002:**
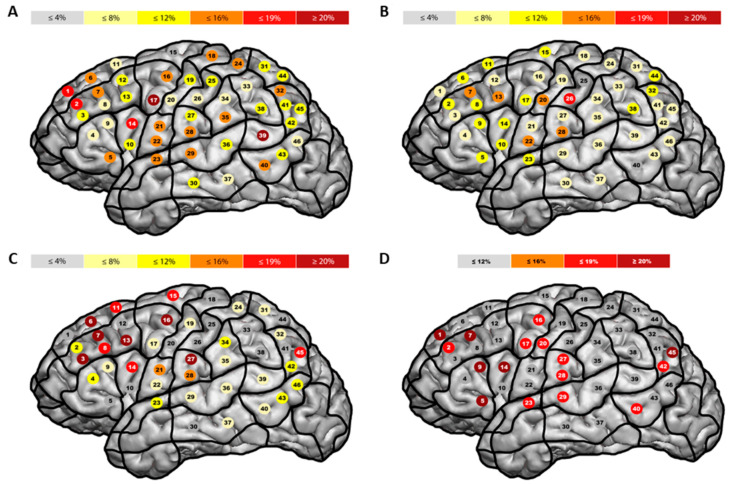
**Mean error rates in all language groups.** The figure shows heat maps of the mean error rates in Balto-Slavic ((**A**), *n* = 24), Indo-European ((**B**), *n* = 25), Arabic ((**C**), *n* = 4), and Chinese languages ((**D**), *n* = 2) based on the raw data.

**Figure 3 brainsci-13-01141-f003:**
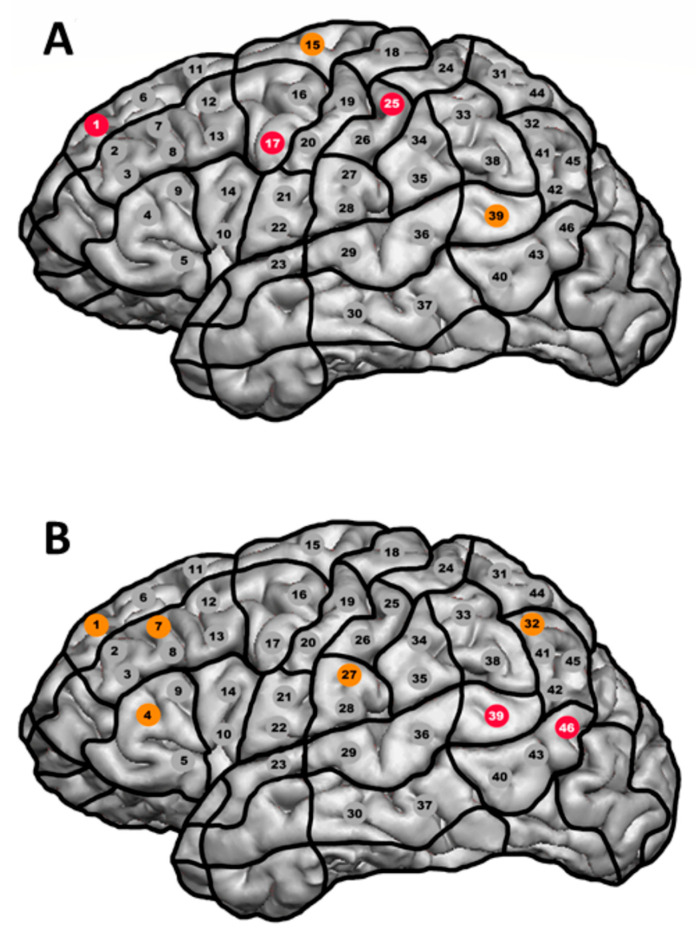
Differences in error rates between (**A**) Balto-Slavic and Indo-European languages and (**B**) the languages spoken by bilingual patients. (**A**) shows significant differences in error rates between Balto-Slavic and Indo-European languages. (**B**) shows significant differences in error rates between the languages spoken by bilingual individuals. Red points mark differences that were significant at *α* = 0.05, while orange points mark differences that were marginally significant at *α* = 0.10.

**Table 1 brainsci-13-01141-t001:** **Patient and mapping characteristics:** The table shows patient characteristics, including age, gender, language, and type of brain lesion (including WHO grade if applicable), and mapping characteristics, including motor threshold, number of stimulations, number of baseline images (the total number of baseline images in brackets), inter-picture interval, and display time.

	Frequency *n* (%)	Mean	SD	Range
Age		44.32	14.99	24–85
Gender				
*Female*	15 (37.50)			
*Male*	25 (62.50)			
Language Groups				
*Indo-European*	33 (52.38)			
*Balto-Slavic*	24 (38.10)			
*Chinese*	2 (3.17)			
*Arabic*	4 (6.35)			
Brain Lesion				
*Arteriovenous Malformation*	5			
*Cavernoma*	4			
*Astrocytoma WHO* ° *I*	3			
*Astrocytoma WHO* ° *II*	3			
*Astrocytoma WHO* ° *III*	7			
*Oligodendroglioma WHO* ° *II*	1			
*Oligodendroglioma WHO* ° *III*	2			
*Glioblastoma WHO* ° *IV*	9			
*Ganglioglioma*	1			
*Metastasis*	4			
*No Operation*	1			
Motor Threshold		33.81	7.69	17–59
Number of Stimulations		143.55	12.07	108–185
Percentage of Baseline Pictures		66.49	17.73	15–100
Inter-Picture Interval		2801.59	341.56	2500–3500
Display Time		933.33	300.26	700–1500

**Table 2 brainsci-13-01141-t002:** **Detailed language and brain lesion characteristics per language group:** The table shows the languages included in the four language groups, as well as the distribution of brain lesion type and location across the language groups.

	Frequency *n* (%)			
	Indo-European (*n* = 33)	Balto-Slavic (*n* = 24)	Arabic (*n* = 4)	Chinese (*n* = 2)
Language				
*Russian*	-	15 (62.50)		
*Croatian*	-	4 (16.66)		
*Serbic*	-	1 (4.17)		
*Polish*	-	1 (4.17)		
*Bulgarian*	-	1 (4.17)		
*Czech*	-	1 (4.17)		
*Lithuanian*	-	1 (4.17)		
*German*	17 (51.52)	-		
*English*	1 (3.03)	-		
*Finnish*	1 (3.03)	-		
*Swedish*	1 (3.03)	-		
*Albanian*	2 (6.06)	-		
*French*	2 (6.06)	-		
*Hungarian*	1 (3.03)	-		
*Romanian*	2 (6.06)	-		
*Portuguese*	1 (3.03)	-		
*Turkish*	1 (3.03)	-		
*Armenic*	2 (6.06)	-		
*Greek*	2 (6.06)	-		
*Arabic*			4 (100)	
*Chinese*				2 (100)
Brain Lesion				
*Arteriovenous Malformation*	5	1	1	-
*Cavernoma*	5	2	-	-
*Astrocytoma WHO I*	1	2	-	-
*Astrocytoma WHO II*	2	3	-	-
*Astrocytoma WHO III*	7	5	1	-
*Oligodendroglioma WHO II*	-	1	-	-
*Oligodendroglioma WHO III*	5	-	-	2
*Glioblastoma WHO IV*	3	7	1	-
*Ganglioglioma*	-	-	1	-
*Metastasis*	5	2	-	-
*No Operation*	-	1	-	-
Brain Lesion Location				
*Left frontal*	11	9	1	2
*Left fronto-temporal*	-	2	-	-
*Left fronto-temoro-insular*	1	1	-	-
*Left temporal*	4	1	1	-
*Left temporo-occipital*	-	1	-	-
*Left parieto-temporal*	-	-	1	-
*Left temporo-insular*	-	2	-	-
*Left temporo-mesial*	-	2	-	-
*Left parietal*	6	1	1	-
*Left occipital*	2	-	-	-
*Left central*	6	-	-	-
*Left insular*	2	2	-	-
*Left corona radiata*	-	1	-	-
*Left intraventricular*	-	1	-	-
*Left limbic*	-	1	-	-
*Bilateral frontal*	1	-	-	-

**Table 3 brainsci-13-01141-t003:** **Languages spoken by bilingual patients:** The table shows the languages spoken by the bilingual patients included in the study.

Patient	Language 1	Language 2	Language 3
1	German	Croatian	-
2	German	Croatian	-
3	German	Croatian	-
4	English	Finnish	Swedish
5	German	Albanian	-
6	German	French	-
7	German	Serbic	-
8	German	Russian	-
9	German	French	-
10	Polish	Russian	-
11	German	Croatian	-
12	German	Albanian	-
13	German	Chinese	-
14	German	Chinese	-
15	German	Portuguese	-
16	German	Czech	-
17	German	Turkish	-
18	German	Greek	-
19	German	Greek	-

**Table 4 brainsci-13-01141-t004:** **Overview of the case numbers per group and analysis:** The table summarizes the case numbers that are included in each language group and in each analysis type. Abbreviation: DTI FT = diffusion tensor imaging fiber tracking.

	Balto-Slavic	Indo-European	Arabic	Chinese	Bilingual Individuals
Error Rates	24	25	4	2	18
DTI FT	23	25	-	-	19

**Table 5 brainsci-13-01141-t005:** **Comparison of white matter pathway volumes between (1) Balto-Slavic and Indo-European languages and (2) languages of bilingual patients**. The table shows the results of the Mann–Whitney U tests comparing the white matter pathway volumes between Balto-Slavic (*n* = 23) and Indo-European languages (*n* = 25) and the results of the Wilcoxon Signed Rank Z tests comparing the white matter pathway volumes between the two languages of bilingual patients (*n* = 19). One asterisk (*) marks differences that were significant at *α* = 0.05. Abbreviations: FA = fractional anisotropy, IFOF = inferior fronto-occipital fascicle, SLF/AF = superior longitudinal fascicle/arcuate fascicle, FAT = frontal aslant tract.

	(1) Balto-Slavic and Indo-European Languages				(2) Languages of Bilingual Patients	
	FA = 0.10		FA = 0.15			
Pathway	*U*	*p*	*U*	*p*	*Z*	*p*
IFOF	284.00	0.942	274.00	0.778	−0.568	0.570
SLF/AF	189.00	0.042 *	226.50	0.206	−0.057	0.955
FAT	241.50	0.248	264.50	0.544	−1.841	0.066
Total	263.00	0.613	287.00	0.992	−0.785	0.433
IFOF %	251.50	0.456	246.50	0.392	−0.544	0.586
SLF/AF %	204.00	0.085	252.00	0.462	−0.827	0.408
FAT %	241.00	0.243	264.00	0.535	−0.365	0.715

## Data Availability

Due to the sensitive character of clinical data, the data included in this manuscript are not publicly available but are available on request from the corresponding author.

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
