# Peer review of "Cortical Location of Language Function May Differ between Languages While White Matter Pathways Are Similar in Brain Lesion Patients"

_brainsci, 2023, doi:10.3390/brainsci13081141_

Round 1

Reviewer 1 Report

The manuscript entitled “Cortical location of language function differs between languages while subcortical pathways are similar in brain lesion patients” by Boerner et al, 2023 demonstrated the underlying subcortical structure might be analogous across languages, there may be differences in the cortical locations of language between various languages as examined by  non-invasive nrTMS language mapping and nrTMS-based DTI FT. The author illustrated the  resection of language-eloquent brain lesions since the location of language cannot be comprehensive across languages or individuals and the cortical location of language function might be plastic whereas the subcortical location is not.

However, some of the points need to be addressed to enhance the importance of the current manuscript:

1. Did the author considers is any other parts of the brain are involved in language function rather than the cortex since astroglioma will affect other regions of the brain?

2. As the author stated in the limitations, reported Cortical differences between Balto-Slavic and Indo-European language groups might be influenced by a cognitive control effect, which must be considered when interpreting the results.

3. Another limitation of the current study is patients with brain lesions imply that cortical plasticity can occur just due to the lesion itself. Furthermore, a brain lesion can persuade tissue displacement, which might bias the subcortical tractography. This should be considered when interpreting the findings of the present study. 

4. The grouping used in the current study not least to gain a decent sample size for statistical comparisons. In addition, future research should investigate whether the variability in language localization between individuals speaking a similar language is lower than the variability within bilingual patients.

Minor editing of the English language is required. 

Author Response

  1. Did the author considers is any other parts of the brain are involved in language function rather than the cortex since astroglioma will affect other regions of the brain?

Answer: Dear reviewer, thank you for your feedback. As stated in the introduction, the aim of the study was to investigate both cortical and subcortical locations of language function. Thus, both cortex and white matter pathways were considered in the analysis. Since brain lesions may impact not only local structures around the lesion but the whole brain, we analyzed language function across the whole left hemisphere and the entire subcortical network regardless of the lesion location (see method sections 2.3 & 2.4). Furthermore, to additionally answer your comment, we added the option to use connectome-based analyses in future studies with the aim to perform a comprehensive network analysis (limitations and conclusions section).

  1. As the author stated in the limitations, reported Cortical differences between Balto-Slavic and Indo-European language groups might be influenced by a cognitive control effect, which must be considered when interpreting the results.

Answer: Dear reviewer, thank you for this comment. Due to several functions of brain regions (e.g., language function as well as cognitive control), cortical differences in these regions between various languages may not reflect language differences alone but may also imply differences in cognitive control. For this reason, we added a paragraph to the limitations section so that readers can take this confounding factor into account when interpreting the results.

  1. Another limitation of the current study is patients with brain lesions imply that cortical plasticity can occur just due to the lesion itself. Furthermore, a brain lesion can persuade tissue displacement, which might bias the subcortical tractography. This should be considered when interpreting the findings of the present study.

Answer: Dear reviewer, thank you for your feedback. To answer your comment, we added this point to the limitations section so that readers can take this point into account when interpreting our results.

  1. The grouping used in the current study not least to gain a decent sample size for statistical comparisons. In addition, future research should investigate whether the variability in language localization between individuals speaking a similar language is lower than the variability within bilingual patients.

Answer: Dear reviewer, thank you for this comment. We know that the language clustering is problematic and difficult from a linguistic point of view. This was also the reason we started the limitations section with this issue. The reason to separate Balto-Slavic from Indo-European languages for the present analysis was not least due to the sample of patients who underwent language-eloquent brain tumor resection at our department and to perform a reliable statistical analysis. We  highlighted your recommendation to investigate differences of individuals speaking similar languages and bilingual individuals.

Comments on the Quality of English Language

Minor editing of the English language is required.

Answer: Dear reviewer, thank you for this comment. We edited the English throughout the manuscript.

Reviewer 2 Report

Dear Authors,

you analyzed "patients have a left-sided perisylvian brain lesion, are right-handed according to the Edinburgh Handedness Inventory" aiming at verifying that "localization of language function differs between various languages". 

In our study there's a methodological major flaw, since you analyze language organization at cortical level (with TMS) and at a subcortical level (with all objective limitation of tractography due to its implicit arbitrariness) in different populations (with all limitations of defining balto-slavic and indo-european that you correctly highlighted) presenting a huge distortion of language network, which is represented by the presence of a lesion in this network. Conclusions cannot be taken as reliable due to this aspect, but misleading. And the sample size is also so poor.

I would consider this study using a population of normals, before going into the domain of dominant peri-sylvian patients (take advantage from the possibility to correlate with pre and post-operative neuropsychological changes).

Best regards

Author Response

Dear Authors,

you analyzed "patients have a left-sided perisylvian brain lesion, are right-handed according to the Edinburgh Handedness Inventory" aiming at verifying that "localization of language function differs between various languages".

In our study there's a methodological major flaw, since you analyze language organization at cortical level (with TMS) and at a subcortical level (with all objective limitation of tractography due to its implicit arbitrariness) in different populations (with all limitations of defining balto-slavic and indo-european that you correctly highlighted) presenting a huge distortion of language network, which is represented by the presence of a lesion in this network. Conclusions cannot be taken as reliable due to this aspect, but misleading. And the sample size is also so poor.

I would consider this study using a population of normals, before going into the domain of dominant peri-sylvian patients (take advantage from the possibility to correlate with pre and post-operative neuropsychological changes).

Best regards

Answer: Dear reviewer, thank you for your feedback. Preoperative nrTMS language mappings of the cortex and nrTMS-based DTI FT of white matter pathways are the standard procedures before undergoing resection of language-eloquent brain lesions in our department. As stated in the introduction, the aim of this analysis was to investigate preoperatively the cortical location of language function and language-eloquent subcortical pathways by exactly these techniques in a brain lesion patient cohort speaking various languages. As discussed in the manuscripts’ limitations section, we highlighted the methodological issues of this cohort and the techniques per se. Additionally, we added a statement on the interpretation of tractography, which was used to make scientific comparisons without showing the true subcortical anatomy.

As stated in the discussion section of the manuscript, there is already literature from our group available on cortical differences between languages in healthy controls (e.g., Tussis et al., 2017, 2019). Based on these findings, we now aimed to investigate cortical and subcortical differences in language function in the patient cohort.

In addition, we know that the language clustering is problematic and difficult from a linguistic point of view. This was also the reason we started the limitations section with this issue.

This paper reports first data and only exploratory analyses and therefore we highlighted in the limitations section points that should be considered by the reader when interpreting the results. In addition, we adjusted the wording in the discussion sections. With this in mind and to answer your comment, we additionally changed the conclusion of our manuscript (limitations and conclusions section).

References:

Tussis, L., Sollmann, N., Boeckh-Behrens, T., Meyer, B., & Krieg, S. M. (2017, May). Identifying cortical first and second language sites via navigated transcranial magnetic stimulation of the left hemisphere in bilinguals. Brain Lang, 168, 106-116. https://doi.org/10.1016/j.bandl.2017.01.011

Tussis, L., Sollmann, N., Boeckh-Behrens, T., Meyer, B., & Krieg, S. M. (2019, Mar 14). The cortical distribution of first and second language in the right hemisphere of bilinguals - an exploratory study by repetitive navigated transcranial magnetic stimulation. Brain Imaging Behav. https://doi.org/10.1007/s11682-019-00082-y

Reviewer 3 Report

This manuscript aimed to study the differences in language-related cortical regions and the underlying anatomical pathway in brain lesion patients who speak different languages using TMS and DTI-based fiber tractography techniques. It's a very interesting topic and well design study. There are some concerns that need to be addressed.

1. In the title and abstract, the term "subcortical pathway" is kind of ambiguous to me, per my understanding, subcortical pathways usually refer to white matter pathways associated with one or more subcortical regions. 

2. Patient of the current study was a group of subjects with left-side perisylvian lesions. The lesion severity will directly affect language performance.  How did the author evaluate the lesion severity? Please add a lesion probability distribution map for the whole patient group or each language group. 

3. The authors used the diffusion tensor model to quantify the local fiber tract orientation and further fiber tractography. Since over half of the white matter regions in the brain may contain crossing fibers which will lead to low anisotropy. The authors compared the fiber volume between different language groups. The crossing fiber issue would introduce biases by the tensor model. The author should at least discuss it and add it as a limitation.

4. The group comparison, there are subjects overlap in different language groups. Did the author try to perform the MannMany Whitney test by removing the overlapped subjects in one subject group?

Minor comments:

1. Section number of Methods is wrong as 1, as well as subsection number of methods were all listed as 2.1.

Author Response

  1. In the title and abstract, the term "subcortical pathway" is kind of ambiguous to me, per my understanding, subcortical pathways usually refer to white matter pathways associated with one or more subcortical regions.

Answer: Dear reviewer, thank you for your feedback. We changed the terminology from “subcortical pathways” to “white matter pathways” in the title, abstract, and throughout the manuscript.   

  1. Patient of the current study was a group of subjects with left-side perisylvian lesions. The lesion severity will directly affect language performance. How did the author evaluate the lesion severity? Please add a lesion probability distribution map for the whole patient group or each language group.

Answer: Dear reviewer, thank you for this comment. Brain lesions included left-sided language-eloquent lesions, and lesion severity was evaluated according to the WHO grading. Detailed information regarding lesion type, severity, and location are summarized in table 2. Therefore, a lesion probability distribution map would double the information in our opinion. In case we correctly understood your comment and a lesions distribution map is still required, we will of course add this map within an additional figure.

  1. The authors used the diffusion tensor model to quantify the local fiber tract orientation and further fiber tractography. Since over half of the white matter regions in the brain may contain crossing fibers which will lead to low anisotropy. The authors compared the fiber volume between different language groups. The crossing fiber issue would introduce biases by the tensor model. The author should at least discuss it and add it as a limitation.

Answer: Dear reviewer, thank you for your feedback. To clarify this point to the readers, we added the following part to the limitations section: “In addition, subcortical tractography was analyzed based on DTI FT and limitations of DTI, e.g., biases due to crossing fibers, need to be considered when interpreting the results. However, DTI sequences with 32 directions and a deterministic algorithm of DTI FT were used to minimize this bias as much as possible.”

  1. The group comparison, there are subjects overlap in different language groups. Did the author try to perform the Mann Whitney test by removing the overlapped subjects in one subject group?

Answer: Dear reviewer, thank you for this comment. Performing tests by removing bilingual subjects from one subject group was not possible since this would have resulted in even smaller sample sizes. As this manuscript describes data gathered during clinical practice, only few patients spoke merely one language (other than German) and comparisons would not have been possible. We highlighted this issue in the limitations section and the conclusion to plan future studies.

Minor comments:

  1. Section number of Methods is wrong as 1, as well as subsection number of methods were all listed as 2.1.

Answer: Dear reviewer, thank you for pointing out this typo. We changed the methods section number to 2.

Round 2

Reviewer 2 Report

Dear Authors,

I appreciate that you face my doubts, but I do not consider the paper suitable for publication for the above mentioned issues

best regards

English is sufficient

Author Response

Dear reviewer,

thank you for the opportunity of a second revision.

As suggested by the editor, we dichotomized the error rates into error/no error and ran a generalized linear mixed model with language and stimulation site as fixed factors and study ID as random factor.

We added the following sentences to the manuscript:  

“Moreover, error rates were dichotomized into “error” and “no error”. Then, we ran a generalized linear mixed model with the dichotomized error variable as target, language and stimulation site as fixed factors, and study ID as random factor.” (part 2.5 Data analysis)

“Analyses using a generalized linear mixed model indicated no significant influence of language (p=1.000) or stimulation site (p=0.807) on the presence or absence of errors.” (parts 3.2 Cortical analysis: error rates and 4.1. Cortical differences and subcortical similarity)

Hereby, we could overcome flaws in the research design (e.g., grouping).

 Moreover, we edited the abstract and the conclusion by adding the following sentences: “These first and exploratory data indicate that the underlying subcortical structure might be similar across languages with initially observed differences in the cortical location of language depending on the semantic processing, while these could not be confirmed by detailed statistical analyses pointing at a similar cortical and subcortical network.” (Abstract)

“The present results indicate that the underlying subcortical structure might be similar across languages, while we initially found differences of error rates in the cortical locations of language between various languages…” and “However, differences in error rates were not significant after correction for multiple comparisons and analyses using a generalized linear mixed model indicated no significant influence of language or stimulation site on the presence or absence of errors. Hence, these detailed statistical analyses could not confirm cortical differences between languages as observed by the pure error rate analysis pointing at a similar cortical and subcortical network.” (part 5. Conclusions)

We hope that we could satisfactorily incorporate the feedback given by you and the other reviewers, which further improved the manuscript.